# Bearing Capacity of Single Pile-Friction Wheel Composite Foundation on Sand-over-Clay Deposit under *V-H-M* Combined Loadings

Yikang Wang [1], Xinjun Zou [1,2,*] and Jianfeng Hu [1]

1  College of Civil Engineering, Hunan University, Changsha 410082, China; wangyk@hnu.edu.cn (Y.W.); hujfeng@hnu.edu.cn (J.H.)
2  Key Laboratory of Building Safety and Energy Efficience of Ministry of Education, College of Civil Engineering, Hunan University, Changsha 410082, China
*  Correspondence: xjzouhd@hnu.edu.cn

**Abstract:** This paper presents numerical modelling to investigate the bearing capacities and failure mechanisms of single pile-friction wheel composite foundation in sand-overlying-clay soil conditions under combined *V-H-M* (vertical-horizontal-moment) loadings. A series of detailed numerical models, with validations of centrifuge testing results, are generated to explore the potential factors influencing the bearing capacity of this composite system. Intensive parametric study is then performed to quantify the influences of the foundation geometry, soil properties, sand layer thickness, pre-vertical loading and lateral loading height on the failure envelopes in the *V-H-M* domain. Last but not least, an empirical design procedure is proposed based on a parametric study to predict the bearing capacity of this composite foundation under various loading conditions, which can provide guidance for its design and application.

**Keywords:** composite foundation; layered soils; numerical modelling; failure envelopes; combined bearing capacity

## 1. Introduction

### 1.1. Concept

Large diameter monopile is often applied to support offshore structures such as wind turbines (OWTs) [1–4]. This is because of its sufficient theoretical basis for analysis design and mature manufacturing technology. However, the bearing capacities provided by monopiles are likely to not be suitable for the next generation of larger wind turbines in more severe environments [5–9]. For example, for OWTs with a full capacity of more than 10 MW, it would be costly and bring more challenges for the construction and installation procedure by merely increasing the diameter and embedment length of the monopile. Thus, a new type of single pile-friction wheel composite foundation has therefore been proposed, which combines the monopile and circular footing foundations. Compared to the traditional monopile foundation, this composite foundation provides higher bearing capacity and stiffness with less construction time and expense [10–12], as is shown in Figure 1. This composite foundation is initiated from the concept of pile caps and embedded retaining walls with stabilizing platforms, in which the addition of the circular wheel increases the shear stress and restoring moment against the lateral deflection [13–15]. The combined wheel can also be applied as an effective reinforcement for existing monopiles to provide additional bearing capacities and protections of the seabed from circum-pile scouring [15–17].

Sand-overlying-clay soil deposit is a complex, but commonly encountered, soil profile in petroleum regions and offshore wind farms, for example, in the Yellow Sea of Korea, North Sea, Gulf of Mexico, South China Sea, offshore India and Thailand [18,19].

Existing understandings only cover the failure mechanisms of conventional types of foundations in such sand-overlying-clay soil profiles [18–22]. For example, Zheng et al. (2018) [23] found a 'punch-through' failure pattern for strip footings in the sand-over-clay layered deposits, where the upper sand block is pushed into the underlying clay due to the vertical compression under *V-H-M* combined loadings. However, the bearing capacities and failure mechanisms of this composite foundation are still not well investigated in such soil conditions. It is also worth noticing that, in order to make full use of the bearing capacity provided by the friction wheel, the composite foundation is suggested to be installed in the regions where the strength of the seabed topsoil is great enough, such as pure sand, sand-over-clay and stiff clay-over-sand soil conditions [24–26]. Hence, a systematic study is badly needed for the application of composite foundations in sand-over-clay layered deposits.

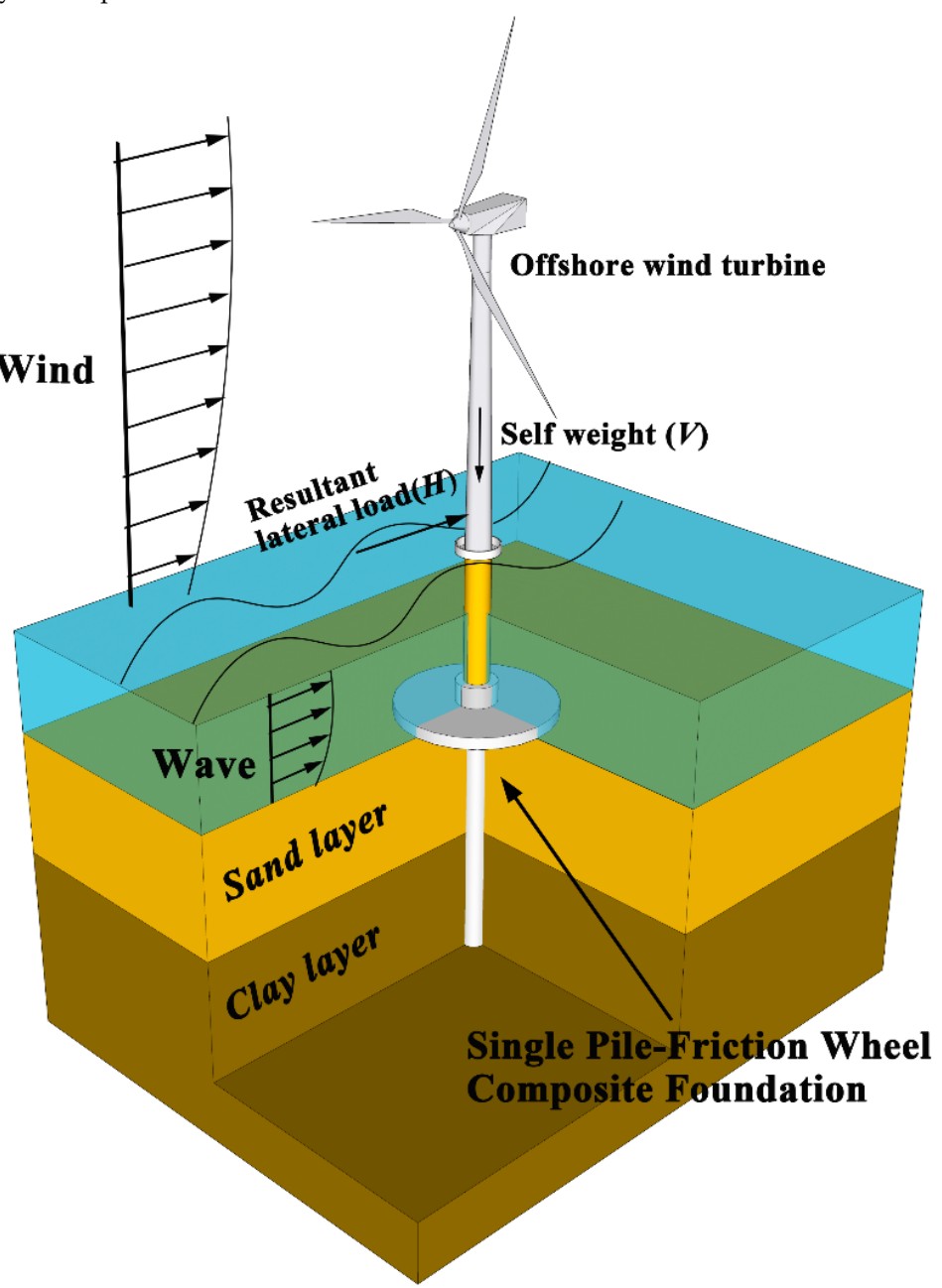

**Figure 1.** Schematic of single pile-friction wheel composite foundation.

*1.2. Previous Work*

A number of studies have been carried out on the bearing capacity of the composite foundation through centrifuge model tests, detailed three-dimensional finite element analyses and theoretical derivations. Most of these studies are only limited to the behaviors of these foundations in pure sand or clay [10,11,24–36].

For the composite system, the interaction effect between the monopile and the wheel has been mostly conducted in two cases: (i) the composite foundation with a wheel diameter ($D_w$) exceeds the pile embedment depth ($L$) ($D_w/L > 1$); and (ii) with a ratio of $D_w/L = 0.5\sim1$. When the wheel diameter is larger than the pile embedment depth as in case (i), Wang et al. (2018) [10] and Yang et al. (2019) [11] conducted centrifuge tests, in which it was found that the lateral bearing capacity of the composite system was actually smaller than the summation of the individual capacities of monopile and wheel. To quantify the reduction, Wang et al. (2018) [10] and Li et al. (2020) [35] proposed an add-up method with a reduction factor to estimate the lateral bearing capacity of the composite foundation. On the contrary, with a smaller wheel as in case (ii), a significant positive interaction effect was found by Lehane et al. (2014) [27] from their centrifuge tests, in which the bearing capacity of the composite foundation is greater than the sum of that provided by the individual monopile and wheel. In addition, Anastasopoulos and Theofilou (2016) [28] proposed that the increase of pile length $L$ had a more pronounced effect on the bearing capacity of the composite foundation than the improvement through increasing wheel diameter $D_w$.

For the failure mechanism of the composite foundation, reported by Anastasopoulos and Theofilou (2016) [28], substantial soil yielding was captured at the edge of the wheel in front of the lateral loading direction. The increase of vertical load ($V$) on the foundation led to the mobilization of the wheel and the soil mass beneath. The load transfer mechanism was also found to be significantly different between two configurations, that is, coupled and decoupled composite systems, respectively. As suggested by Stone et al. (2018) [36], for a coupled system, the wheel rotated as a rigid body with the center on the pile axis; however, it was found that the uniform settlement of the wheel triggered by the vertical load played a dominant role in the failure mechanism for the decoupled arrangement. In order to assess the bearing capacity of the composite foundation, an empirical design procedure was proposed by adopting the concept of the soil wedge under the wheel in front of the composite foundation [34]. To investigate the bearing capacities of the composite foundation under *V-H-M* combined loadings, centrifuge tests have been carried out by El-Marassi (2011) [25] and Yang et al. (2019) [11]. It was concluded that the lateral load bearing capacity and the moment resistant capacity were improved by increasing the vertical load $V$ on the composite foundation, and the maximum lateral load resistance occurred at $V/V_{ult} = 0.4\sim0.5$ ($V_{ult}$ is the ultimate vertical loading capacity).

Unlike the previous studies, which mainly focus on the bearing behavior of the composite foundation in pure sand or clay, in this study detailed finite element analyses are carried out to study the bearing capacities and failure mechanisms of the composite foundation in sand-overlying-clay deposits under *V-H-M* combined loadings. Besides that, an empirical design method is innovatively proposed based on an intensive parametric study to quantify the failure envelopes considering the influences of foundation geometry ($D_w/L$ and $L/D_p$: $D_w$ is the wheel diameter; $D_p$ is the wheel diameter; $L$ is the pile embedment depth), soil layer distribution condition ($T_s/L$: $T_s$ is the thickness of sand layer), soil properties ($\tan(\varphi)\gamma'_s D_p/s_{um}$: $\varphi$ is the internal friction wheel of sand, $\gamma'_s$ is the buoyant unit weight of sand, $s_{um}$ is the undrained shear strength of clay at the soil surface) and the lateral loading height ($e$), which are essential for design management during application.

## 2. Materials and Methods

Detailed three-dimensional finite element analyses are conducted to examine the failure mechanism and the bearing capacity of the composite foundation in sand-overlying-clay using commercial software ABAQUS [37]. The numerical modeling helps to investigate

the soil deformation and stress distribution around the composite foundation, and the failure surface in the limiting state.

### 2.1. Finite Element Model

As shown in Figure 2, the soil domain is set with the horizontal boundary width (horizontal distance from the wheel edge to the mesh boundary) as 5 $D_w$ and the vertical boundary height (vertical distance from the pile end to the mesh boundary) as 2 $L$, which are large enough to ensure minimal boundary effects [30–32]. Three-dimensional 8-node linear brick elements (C3D8R) with reduced integration are adopted in the models. Relatively fine meshes are adopted around the foundation for a higher computational accuracy, and coarser mesh sizes are employed in the soil away from the foundation to optimizing the computer resources needed. Hinge supports are applied at the bottom of the model and roller conditions are set along the vertical sides of the soil domain. The vertical boundary in the front side of the model is fixed in the horizontal direction (y direction) and set free in the other two directions (x and z directions). In order to simplify the simulation, the analyses involve a "wished in place" pile installation before loading. Beam elements are used to model the tower structure with a linear elastic material property. A lumped mass is adopted to simulate the super-structure assembly, which applies the vertical pre-load by its self-weight [38,39]. The corresponding load can be extracted from the loading point as shown in Figure 2. The resultant lateral load ($H$) is applied at a certain height ($e$) on the tower above the mudline, which leads to the combined lateral load ($H$) and the designed moment ($M = H \times e$) acting on the foundation. The height of the lateral load was determined according to the engineering practice [40]. The lateral load is applied in the form of continuous lateral displacement until the foundation reaches its ultimate bearing state, which is determined by the tangent intersection method [41,42]: the load corresponding to the intersection point of the lateral load-displacement curve is regarded as the ultimate bearing capacity of the foundation.

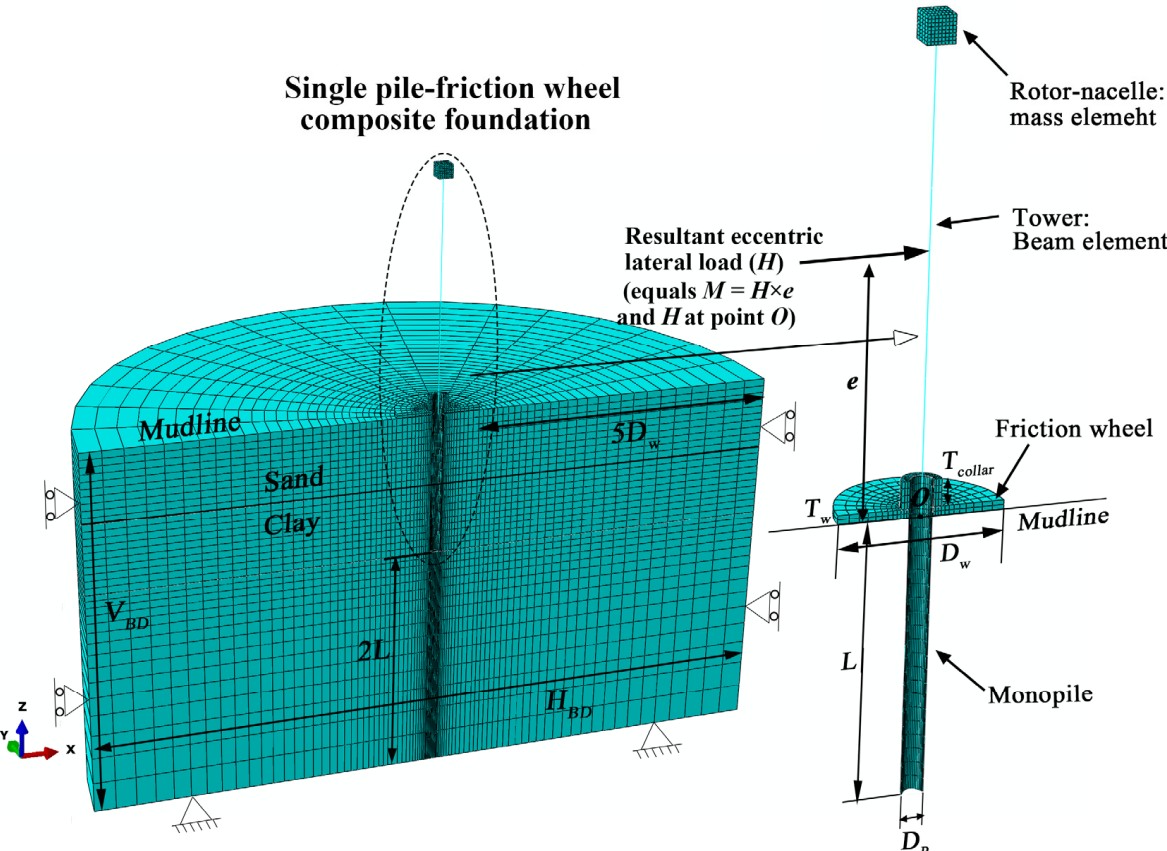

**Figure 2.** Geometry and mesh of the single pile-friction wheel composite foundation models.

### 2.2. Constitutive Model and Soil Properties

In this study, a composite system used to support a wind turbine is considered, which consists of a hollow circular pipe pile with pile diameter $D_p$ = 4 m, wall thickness $t$ = 0.2 m and a solid wheel with the thickness $T_w$ = 2 m. The wheel diameter $D_w$ and pile embedment depth $L$ are determined as ratios of $D_w/L$ and $L/D_p$ varying from 0 to 0.95 and 7 to 13, and the effects of wheel thickness ($T_w$) and height of wheel collar ($T_{collar}$) are also investigated. The steel pile and wheel are modelled as an elastic-perfectly material with the Young's modulus of 210 GPa, Poisson's ratio $\nu$ of 0.17, and unit weight of 78.5 kN/m$^3$. This assumption of the foundation has also been widely used in many other researches [11,28,31,32], and good predictions of the bearing performance have been obtained compared to that of the real structure. The sand is assumed to have the buoyant unit weight $\gamma'_s$ of 9.8 kN/m$^3$, internal friction angle $\varphi$ of 33°, Poisson's ratio ($\nu$) of 0.3, Young's modulus $E_s$ of 40 MPa, dilation angle $\psi$ of 3°, and cohesive force $c$ of zero, which are obtained from the triaxial tests. The friction coefficient between the sand and foundation is measured as 0.3 from the lab tests as reference [32]. By adopting a perfect contact with Coulomb's tangent friction model, interactions between the foundation and the sand are simulated allowing detachment and sliding. The buoyant unit weight of clay $\gamma'_c$ is taken as 8.5 kN/m$^3$ [32]. A uniform stiffness ratio $E_c/s_u$ = 500 (where $E_c$ is the Young's modulus of the clay, $s_u$ is the undrained shear strength of clay) [43] is adopted throughout the soil domain and the Poisson's ratio $\nu$ is set as 0.49 to model the fully undrained condition. The undrained shear strength of clay is described with a normalized soil strength gradient in a linearly increasing strength profile, as follows:

$$s_u = s_{um} + kz, \tag{1}$$

where $s_{um}$ is the undrained shear strength at soil surface, $k$ is the shear strength gradient with the submerged depth $z$ and is taken as 1.75 kPa/m in the following analyses [4,44]. The shear strength of the nodal joint elements near the clay/pile interface is limited to $\alpha s_u$ ($\alpha$ is the interface friction coefficient which is set as 0.3 for the composite foundation/clay interface). This method has been successfully adopted in simulating the interaction between pile and clay in small strain analyses, where good predictions were obtained [45,46]. The gravity is initially applied to the soil to establish geostatic stresses state in soils, and the coefficient of lateral earth pressure at rest is set as $K_0 = 1 - \sin\varphi$.

### 2.3. Numerical Model Validation

The generated numerical model is validated using the lateral load-displacement curves that are obtained from the centrifuge tests [27]. (Case 1: $D_w/L$ = 0.5, $V$ = 5.3 MN; Case 2: wheel only, $V$ = 5.9 MN; Case 3: wheel only, $V$ = 12.7 MN; Case 4: $D_w/L$ = 0.5, $V$ = 6.4 MN; Case 5: $D_w/L$ = 0.5, $V$ = 13.2 MN, where the other parameters are: $D_p$ = 3.33 m, $e/D_p$ = 9, $T_s/L$ = 1; Group I, Table 1). As shown in Figure 3, reasonably close agreements can be found between the results obtained from finite element method (FEM) and the centrifuge tests. We can therefore be confident that the above developed numerical modeling method could reasonably closely replicate the bearing behavior of the composite foundation in sand-overlying-clay deposit.

<div align="center">Table 1. Summary of the numerical simulations.</div>

| Analysis | $D_w/L$ | $V/V_{ult}$ | $e/D_p$ | $T_s/L$ | $s_{um}$ | $\varphi$ (sand) | $T_w/D_p$ | Notes |
|---|---|---|---|---|---|---|---|---|
| Group I | 0, 0.5, wheel | 5.9, 12.7, 6.4, 13.2 (MN) | 9 | 1 | 0 | 33° | 0.8 | Numerical model validation [27] |
| Group II | 0~0.8 | 0 | 10 | 0.5 | 30 kPa | 33° | 0.5 | Investigation for the effect of the wheel diameter |
| Group III | 0.6 | 0~95% | 6~16 | 0.5 | 30 kPa | 33° | 0.5 | Investigation for the effect of the combined pre-vertical load and moment |
| Group IV | 0.6 | 0 | 10 | 0.1~0.8 | 20 kPa, 40 kPa ($k = 1.75$ and 2 kPa) | 20~40° | 0.5 | Investigation for the effect of thickness of sand layer |
| Group V | 0.6 | 0 | 10 | 0.5 | 30 kPa | 33° | 0.5~1.25 | Investigation for the effect of wheel thickness |

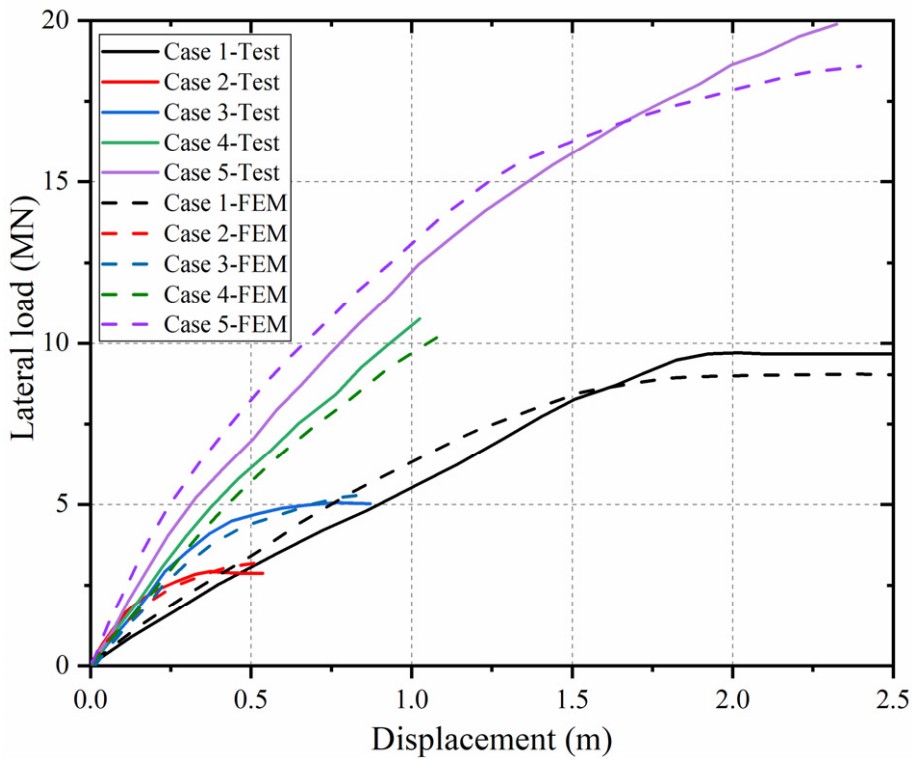

**Figure 3.** Lateral load-deflection curves obtained from FEM and centrifuge tests.

## 3. Effect of Different Influencing Factors

### 3.1. Effects of Wheel Diameter and Pile Embedment Depth ($D_w/L$ and $L/D_p$)

To investigate the effects of the wheel diameter and pile embedment depth on the lateral bearing performance, a group of numerical analyses is carried out with seven different wheel diameter ratios for the composite foundation ($D_w/L = 0$, 0.3, 0.4, 0.5, 0.6, 0.7 and 0.8, $L/D_p = 7$, 9, 11 and 13; Group I, Table 1). Figure 4a shows the normalized lateral capacities ($H_{ult}/(2\pi R^3 \gamma'_s)$) of the composite foundation with different wheel diameters. It is apparent that the lateral capacity and the wheel diameter show an almost linear relationship when $D_w/L$ is within the range of 0 to 0.5. The slope of the curve experiences a rapid increase at the turning point of about $D_w/L = 0.5$, which shows a good agreement with the observation from the previous investigations [10,11,32]. The soil plastic strain distribution at the ultimate condition in terms of plastic strain within and around the composite

foundation are also illustrated in Figure 4a. It can be seen that through comparing the composite foundation and the monopile, the main difference of the failure pattern lay on the area at the right side of the foundation. When $D_w/L = 0$, the plastic strain generates from the bed surface at about $3 D_p$ away from the right side of the pile, which propagates inwards to the right exterior side of the pile, forming a hill-type failure zone ($f_2$). Besides, the failure zone $f_3$ develops from the mudline to the interface of the sand and clay layer along the left pile body side. The failure zone $f_4$ occurs on both sides of the pile in the clay layer due to the rotational movement of the pile. As for the composite foundation with $D_w/L = 0.6$, a semi ellipse-shape failure zone ($f_1$) occurs near the right external side of the wheel edge due to the intrusion of the wheel during rotation. Failure zone ($f_2$) at the intersection area of the pile and the wheel shows a relative smaller value compared to that of the monopile. The failure zone ($f_3$) propagates along the left side of the pile, and eventually reaches the interface of the two soil layers. Similar shape of failure zone ($f_4$) can be captured on the composite foundation as that on the monopile. It is evident that the failure pattern of the composite foundation is considerably different from that of the monopile due to the integration of the wheel. As the wheel diameter $D_w/L$ increases from 0.6 to 0.8, the failure zone ($f_1$) enlarges along the loading direction, indicating a greater bearing capacity.

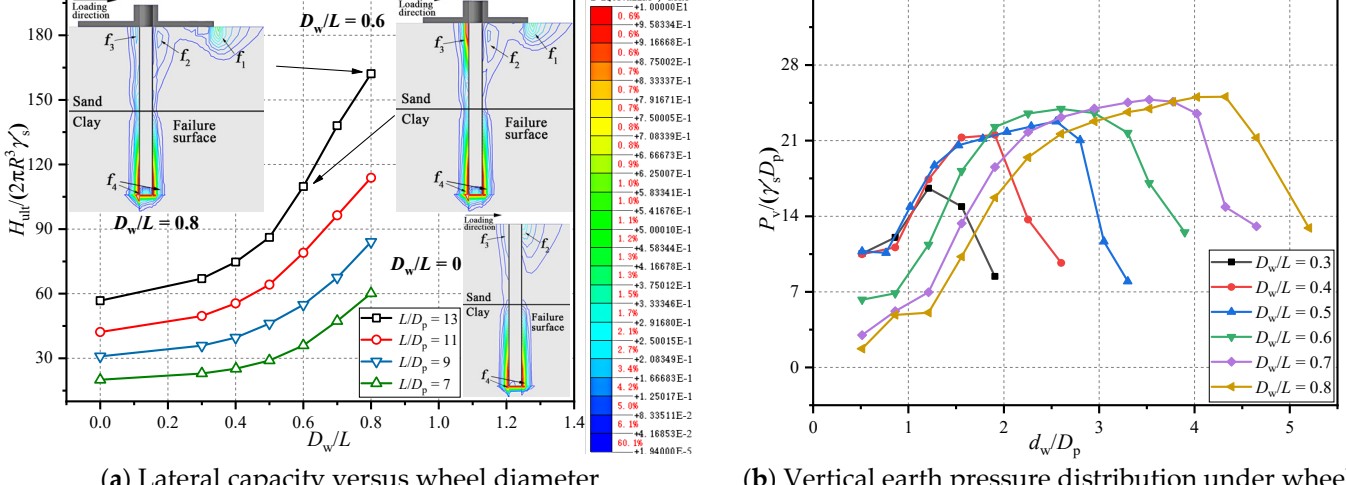

(**a**) Lateral capacity versus wheel diameter      (**b**) Vertical earth pressure distribution under wheel

**Figure 4.** Effect of wheel diameter on the lateral capacity and the failure mechanism.

Figure 4b shows the vertical passive earth pressure distribution under the wheel through presenting the normalized passive pressure ($P_v/(\gamma'_s D_p)$) versus normalized distance to the wheel center ($d_w/D_p$) relationship. It is evident that the overall distribution of the vertical passive earth pressure is similar for the foundation with different wheel diameters, which indicates an increasing trend as the distance develops towards the wheel edge with the maximum value at about $0.75 D_w$. It can be explained that the right intersection soil area between the pile and the wheel experiences soil densification due to the compression of the pile and the wheel, which generates a higher soil resistance vertically. The total vertical passive resistance generated under the wheel increases as the wheel diameter increases, which provides a larger bearing area.

### 3.2. Effects of Pre-Vertical Load and Lateral Loading Height ($V/V_{ult}$ and $e/D_p$)

Numerical modeling is carried out to examine the effect of the pre-vertical load and the lateral loading height, where the foundation geometry and the soil properties are listed in Group III of Table 1. The relationship of the normalized lateral bearing capacity and the pre-vertical load is plotted in Figure 5. It can be found that, with $D_w/L = 0.6$, the maximum lateral bearing capacity of the composite foundation occurs when $V/V_{ult} = 60\%$

for different lateral loading heights. It can be concluded that the lateral loading height has little influence on the shape of the failure envelopes but only alter asymptotes.

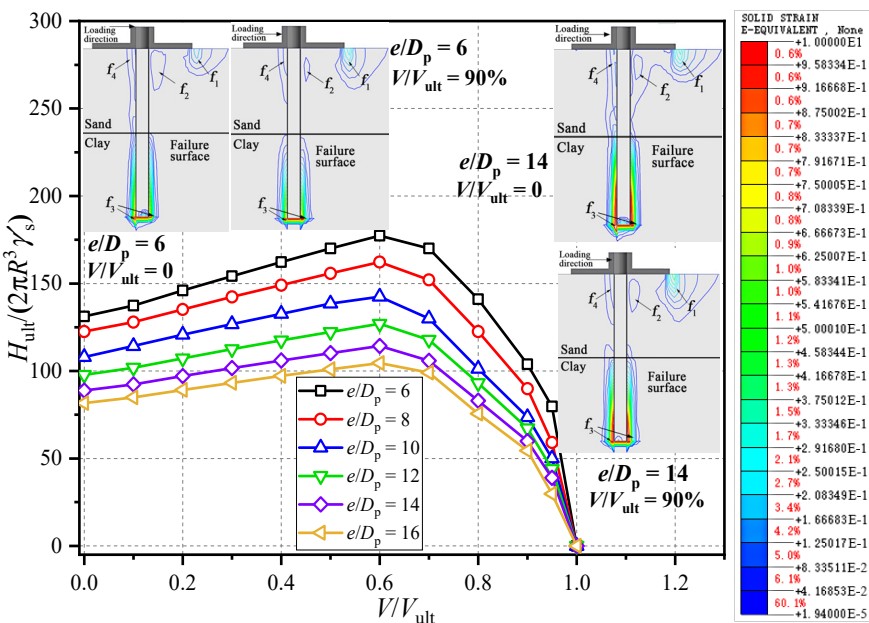

**Figure 5.** Effect of the pre-vertical loading on the lateral capacity and the failure mechanism with different loading height.

Figure 5 also illustrates the failure mechanism of the composite foundation under the combined *V-H-M* loadings. The plastic strain area $f_1$ is found to expand downwards as the pre-vertical load increases, which means more soils under the wheel are mobilized to provide the bearing capacity. In addition, with the elevation of the lateral loading height, plastic strain area $f_1$ and $f_3$ increase and show greater values. This behavior can be attributed to the increasing dominance of the rotational movement of the composite foundation as the lateral loading height increases, which results in a lower initial stiffness and lateral bearing capacity of the composite foundation.

### 3.3. Effects of Sand Layer Thickness and Soil Properties ($s_{um}$, k and φ)

To study the effect of the soil layer thickness and the soil properties, a group of numerical analyses is conducted with $T_s/L = 0.1{\sim}0.8$, $s_{um} = 20$ kPa and 40 kPa and $k = 1.75$ and 2 kPa/m (Group IV, Table 1). The influences of the internal friction angle on sand (φ) and $T_s/L$ are presented in Figure 6, where the normalized lateral bearing capacity is plotted against $T_s/L$ with different φ and $s_{um}$. It can be found that the lateral bearing capacity of the composite foundation increases nonlinearly as $T_s/L$ increases. The increasing trend becomes more evident with a greater value of φ, which is partly due to the *P*-Δ effect. From the failure mechanism of the composite foundation depicted in Figure 6a,b, it can be seen that, when $T_s/L = 0.1$ and φ = 20°, a "punching-through" failure pattern is developed in region ($f_1$), which includes a rigid block near the pile beneath the footing, a passive wedge at lower right of the wheel in the sand layer, and a fan in the lower clay layer. Similar conclusions are also reported by Zheng et al. (2018) [23] in their investigation of the failure mechanism of the strip footing in sand-over-clay soil deposit. In addition, more evident plastic strain areas ($f_2$) and ($f_3$) at the interface of the two soil layers and the plastic strain areas ($f_4$) under the wheel are captured when $T_s/L = 0.1$ and φ = 20° as compared to that of φ = 40°. When $T_s/L$ increases to 0.8, a significant plastic strain area ($f_4$) can be seen on the lower left region of the wheel when φ = 20°. Generally, increasing $s_{um}$ can significantly enhance the bearing capacity of the composite foundation when $T_s/L$ is relatively small, while soil strength gradient *k* has limited influence on the bearing capacity, which can be concluded by the $H_{ult}/(2\pi R^3 \gamma'_s)$-$T_s/L$ curves and the failure mechanisms.

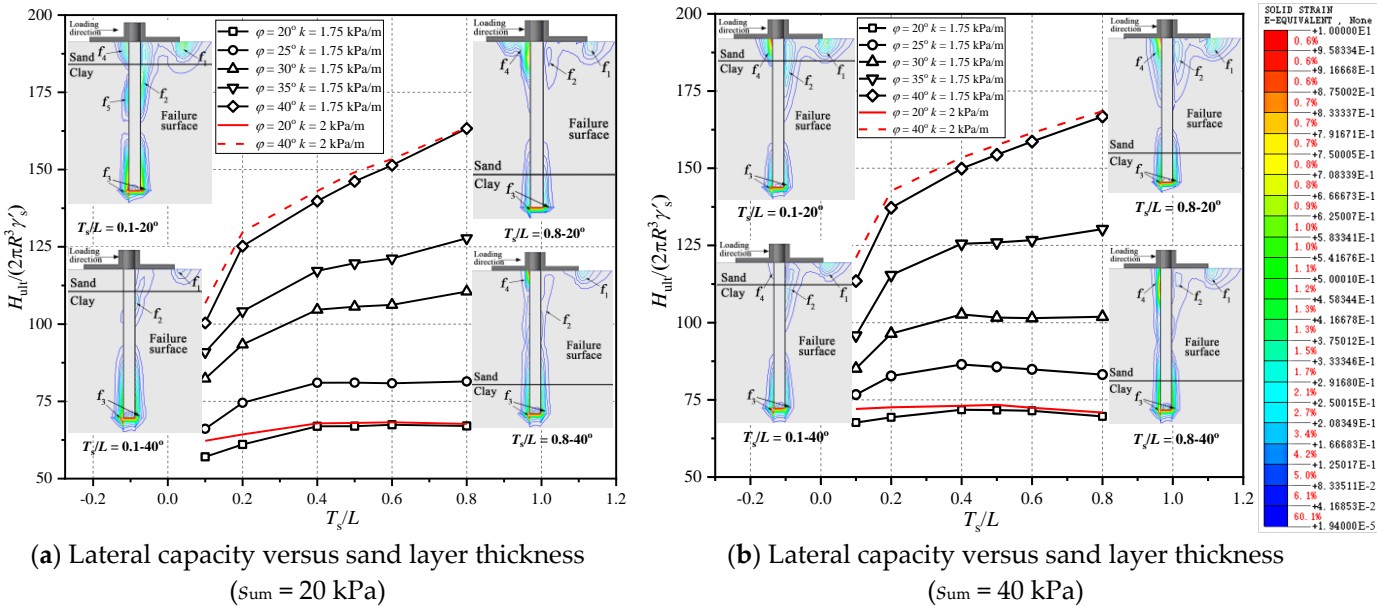

(**a**) Lateral capacity versus sand layer thickness ($s_{um}$ = 20 kPa)

(**b**) Lateral capacity versus sand layer thickness ($s_{um}$ = 40 kPa)

**Figure 6.** Effect of wheel diameter on the lateral capacity and the failure mechanism.

Figure 7 demonstrates the lateral earth pressure distributions along the right side of the pile body under different $T_s/L$ and $\varphi$ by plotting normalized lateral earth pressure $P_p/(2\pi R\gamma'_s)$ versus depth below the mudline $z/D_p$. In terms of $T_s/L = 0.1$, there are no obvious differences between the earth pressure distributions when $\varphi = 20°$ and $40°$, with the same depth of maximum earth pressure $z_{max} = 0.4\,L$, which is in accordance with the case when the composite foundation is in pure clay [32]. Besides that, no abrupt change is captured at $T_s/L = 0.1$, and it indicates that the bearing capacity is mainly controlled by the combined wheel rather than the soil resistance provided by the pile. For $T_s/L = 0.4$, the overall earth pressure distribution is significantly different from that of $T_s/L = 0.1$, with its $z_{max}$ elevate to above $0.3\,L$. Apparently, a sharp decrease of soil resistance can be found at $T_s/L = 0.4$ as the depth descends. In addition, the depth of the maximum earth pressure raises with larger $\varphi$ ($\varphi = 20°$: $z_{max}/L = 0.28$; $\varphi = 40°$: $z_{max}/L = 0.19$). As $T_s/L$ increases from 0.4 to 0.8, for $\varphi = 20°$, the earth pressure along pile barely changes, while it experiences an obvious increase within the range of $z/L = 0\sim0.55$ when $\varphi = 40°$, which is consistent with the phenomenon shown in Figure 6.

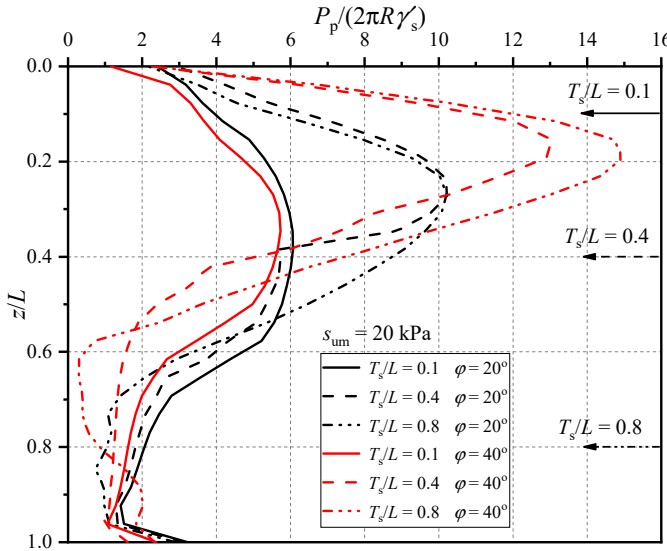

**Figure 7.** Lateral soil resistance along the pile under different sand layer thicknesses.

*3.4. Effects of Wheel Thickness and Height of Collar ($T_w/D_p$ and $T_{collar}/D_p$)*

To explore the effect of wheel thickness $T_w$ and the height of collar $T_{collar}$, a group of analyses was conducted (Group V, Table 1). Figure 8 illustrates the relationship between ultimate lateral bearing capacities and normalized wheel thickness $T_w/D_p$ with different $T_{collar}/D_p$. It shows that $T_{collar}$ has a negligible influence on the capacity of composite system. As $T_w/D_p$ changes from 0.5 to 1.25, the bearing capacity increases linearly by about 17%. This is because that additional vertical pressure is applied to the soil underneath the wheel when $T_w$ is greater, and it can be validated by the plastic strain distribution displayed in Figure 8, which is similar to the situation in Figure 5 when vertical load is applied to the composite system. In addition, with the aim of cost-efficiency in the manufacture and installation of the composite foundation, a friction wheel with a greater thickness would not be suitable for the application of this type of foundation. Therefore, based on the above analysis, it can be confirmed that the $T_w/D_p$ and $T_{collar}/D_p$ have minimal influences on the bearing capacity.

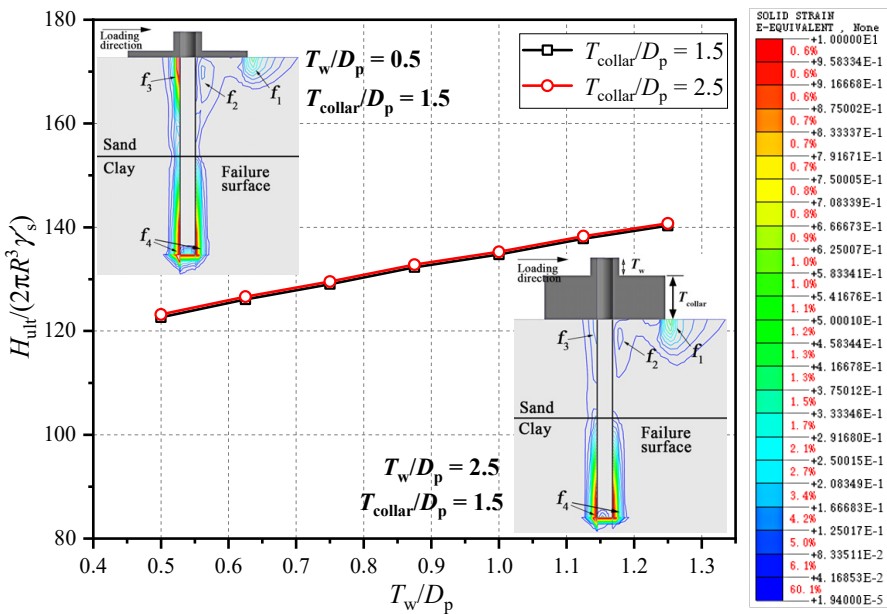

**Figure 8.** Effect of wheel thickness and height of wheel collar.

## 4. Failure Envelopes in the *V-H-M* Domain

### 4.1. Parametric Study

Parametric studies are carried out to quantify the influences of the soil properties $(\tan(\varphi)\gamma'_s D_p/s_{um})$, sand layer thickness $(T_s/L)$ and foundation geometry $(D_w D_p/L^2)$ on the bearing capacity under combined *V-H-M* loadings. Figure 9 presents the influence of soil properties on the bearing capacities in the *V-H-M* domain by plotting failure envelopes with varying $\tan(\varphi)\gamma'_s D_p/s_{um}$. ($D_w D_p/L^2 = 0.062$ and 0.08, $T_s/L = 0.1$ and 0.7, $e/D_p = 15$). Figure 9a shows that, the shape and value of failure envelopes are both related to the soil properties. From Figure 9b, it can be observed that the peak lateral capacity increases at an accelerating rate with the increase of $\tan(\varphi)\gamma'_s D_p/s_{um}$. Figure 10 shows the effect of the foundation geometry on the failure envelops in the *V-H-M* domain. The failure envelopes expand with the increase of $D_w D_p/L^2$ under a constant upper sand thickness, which means the foundation geometry has little influence on the failure mechanism but affects the magnitude of the bearing capacity of the composite foundation. As can be seen in Figure 11, when $T_s/L = 0.1$, under a constant $D_w D_p/L^2$, the lateral bearing capacity decreases continuously when the vertical load increases. However, as $T_s/L$ exceeds 0.1, the peak bearing capacity can be observed in the failure envelopes. The normalized vertical load corresponding to the peak value grows as the thickness of the upper sand layer increases. For instance, it increases from 50 to 180 as $T_s/L$ increases from 0.3 to 0.7 when

$D_\mathrm{w}D_\mathrm{p}/L^2 = 0.062$. This is because a thin layer of top sand may result in a different failure pattern like the "punch-through" failure under the combined *V-H-M* capacities.

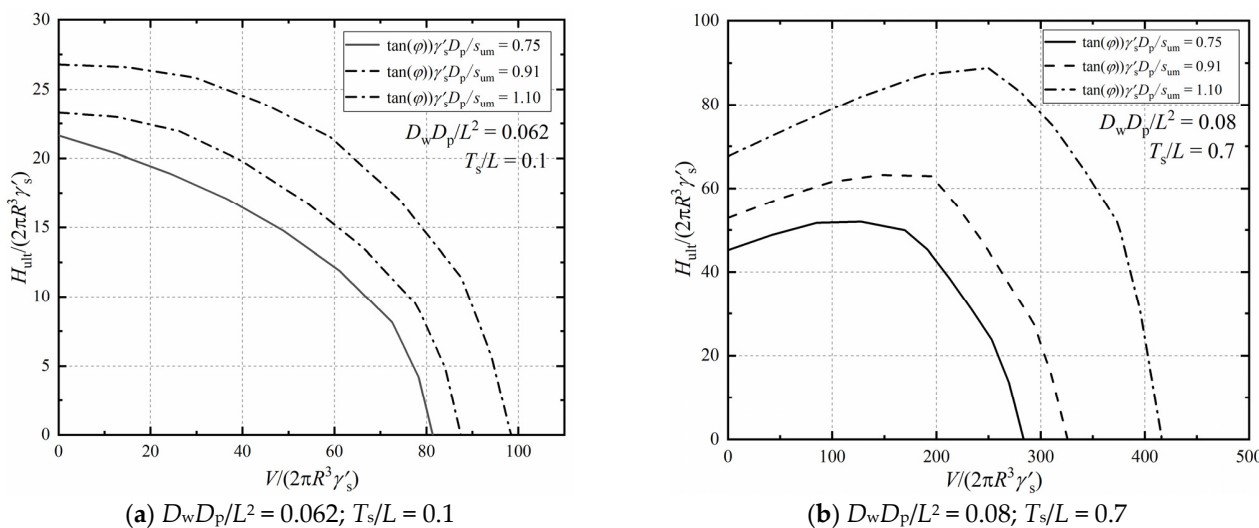

**(a)** $D_\mathrm{w}D_\mathrm{p}/L^2 = 0.062$; $T_\mathrm{s}/L = 0.1$　　　　　　　**(b)** $D_\mathrm{w}D_\mathrm{p}/L^2 = 0.08$; $T_\mathrm{s}/L = 0.7$

**Figure 9.** Effects of soil properties ($\tan(\varphi)\gamma'_\mathrm{s}D_\mathrm{p}/s_\mathrm{um}$).

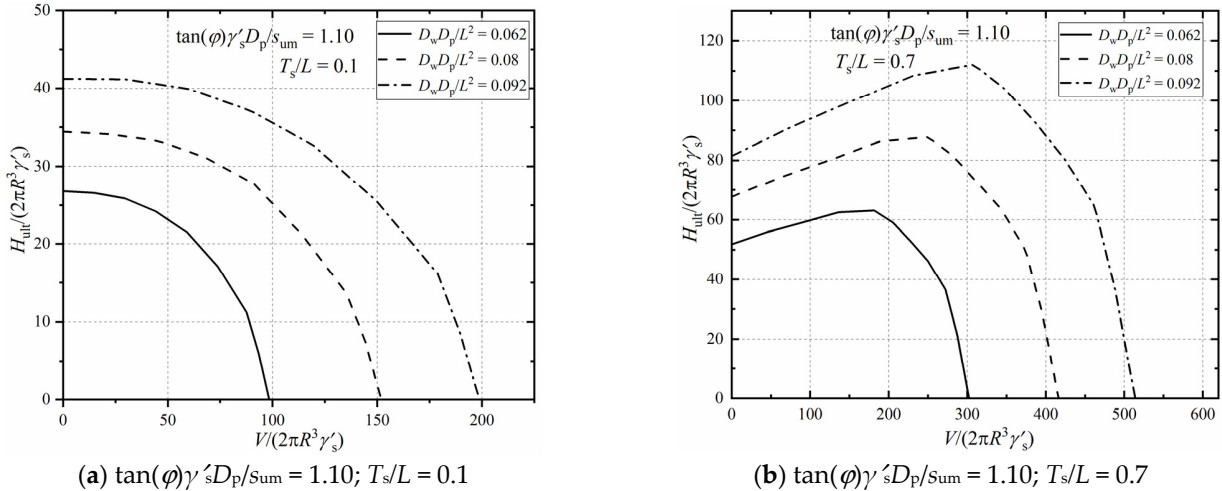

**(a)** $\tan(\varphi)\gamma'_\mathrm{s}D_\mathrm{p}/s_\mathrm{um} = 1.10$; $T_\mathrm{s}/L = 0.1$　　　**(b)** $\tan(\varphi)\gamma'_\mathrm{s}D_\mathrm{p}/s_\mathrm{um} = 1.10$; $T_\mathrm{s}/L = 0.7$

**Figure 10.** Effects of wheel diameter and pile embedment depth ($D_\mathrm{w}D_\mathrm{p}/L^2$).

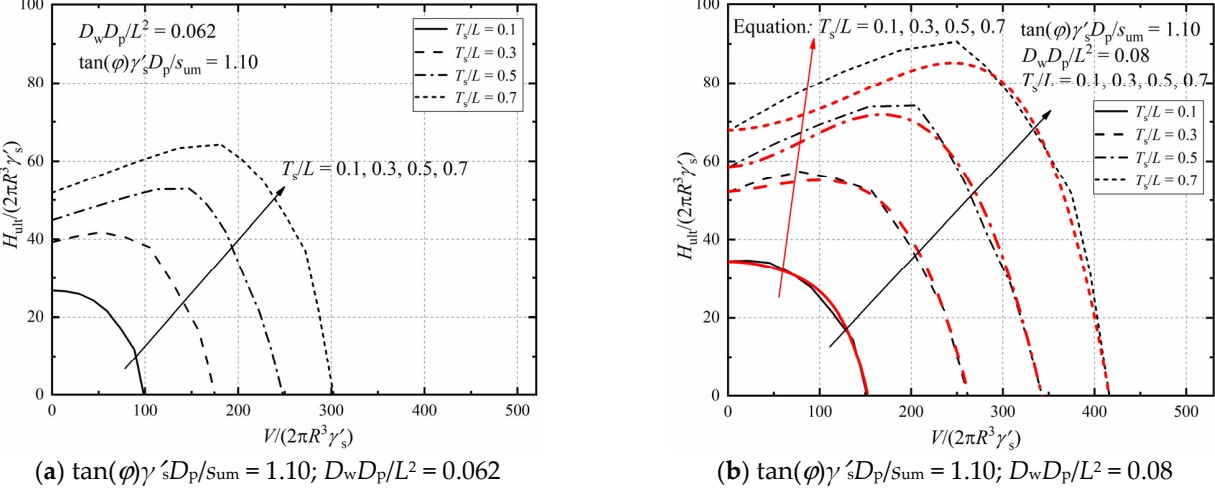

**(a)** $\tan(\varphi)\gamma'_\mathrm{s}D_\mathrm{p}/s_\mathrm{um} = 1.10$; $D_\mathrm{w}D_\mathrm{p}/L^2 = 0.062$　　　**(b)** $\tan(\varphi)\gamma'_\mathrm{s}D_\mathrm{p}/s_\mathrm{um} = 1.10$; $D_\mathrm{w}D_\mathrm{p}/L^2 = 0.08$

**Figure 11.** Effects of sand layer thickness ($T_\mathrm{s}/L$).

### 4.2. Approximate Expression of the Bearing Capacity in the V-H-M Domain

Based on the parametric study, it can be concluded that both the sand layer thickness $T_s/L$ and the soil properties $\tan(\varphi)\gamma'_s D_p/s_{um}$ influence the shape of failure envelopes under the combined *V-H-M* loadings. To estimate the bearing capacities conveniently, the failure envelopes normalized by the corresponding $H_0$ and $V_{ult}$ are shown in Figure 12 ($H_0$ is the uniaxial horizontal capacity; $V_{ult}$ is the vertical capacity of the foundation), and consider the mobilization of the sand layer thickness and the soil properties. Inheriting the format of the empirical formula by El-Marassi (2011) for composite foundations in pure sand, Equation (2) is proposed to determine the shape of failure envelopes in the combined *V-H-M* domain. *A*, *B*, *C*, *D*, and *E* are coefficients related to $T_s/L$ and $\tan(\varphi)\gamma'_s D_p/s_{um}$, which are derived in Table 2. ($h_0$ is the dimensionless uniaxial horizontal capacity; $v_0$ is the dimensionless vertical load)

$$\frac{H_{ult}}{H_0} - A \times \left(\frac{V}{V_{ult}}\right)^B \times \left(C - D \times \left(\frac{V}{V_{ult}}\right)^E\right) = 1 \tag{2}$$

$$h_0 = a_1 + b_1 \times D_{w0} + (c_1 + d_1 \times D_{w0}) \times e_0 + (e_1 + f_1 \times D_{w0} + (g_1 + h_1 \times D_{w0}) \times e_0) \times \varphi_0 \tag{3}$$

$$v_0 = a_2 + b_2 \times D_{w0} + (c_2 + d_2 \times D_{w0}) \times \varphi_0 \tag{4}$$

$$h_0 = \frac{H_0}{2\pi R^3 \gamma'_s}, \quad v_0 = \frac{V_{ult}}{2\pi R^3 \gamma'_s}, \quad D_{w0} = \frac{D_w D_p}{L^2}, \quad \varphi_0 = \frac{\tan(\varphi)\gamma'_s D_p}{s_{um}}, \quad e_0 = \frac{e}{D_p}.$$

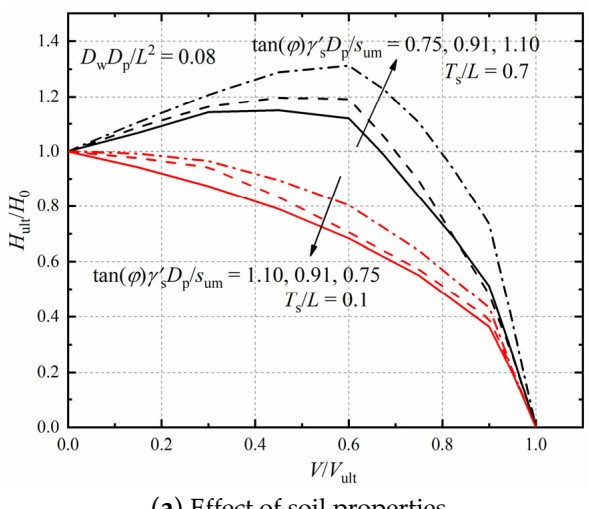

(**a**) Effect of soil properties

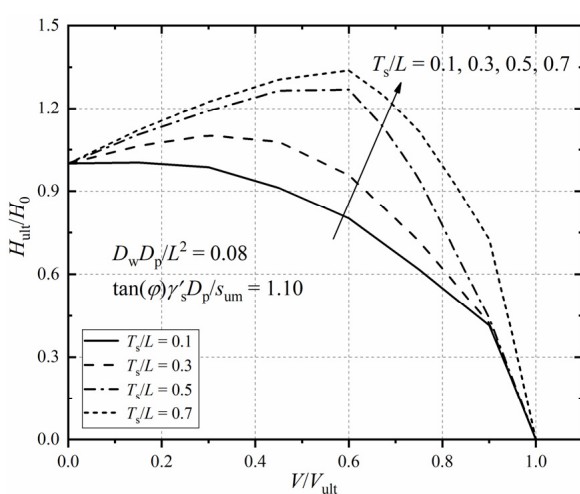

(**b**) Effect of sand layer thickness

**Figure 12.** Typical FE results of the normalized bearing capacity in the *V-H-M* domain.

**Table 2.** Values of *A*, *B*, *C*, *D* and *E*.

| $T_s/L$ | $\tan(\varphi)\gamma'_s D_p/s_{um}$ | *A* | *B* | *C* | *D* | *E* |
|---|---|---|---|---|---|---|
| 0.1 | 0.754 | 1 | 1.177 | −0.505 | 0.439 | 3.784 |
| 0.1 | 0.848 | 1 | 1.847 | −0.790 | 0.209 | 27.125 |
| 0.1 | 0.949 | 1 | 2.200 | −0.790 | 0.210 | 45 |
| 0.1 | 1.096 | 0.739 | 1.913 | −0.551 | 0.774 | 4.49 |
| 0.3 | 0.754 | 1 | 2.172 | 0.508 | 1.411 | 0.194 |
| 0.3 | 0.848 | 1 | 2.8 | 2.725 | 3.659 | 0.201 |
| 0.3 | 0.949 | 1 | 1.85 | 0.351 | 1.307 | 1.55 |
| 0.3 | 1.096 | 1 | 1.355 | 0.322 | 1.314 | 2.552 |
| 0.5 | 0.754 | 1 | 1.256 | 0.914 | 1.653 | 0.891 |
| 0.5 | 0.848 | 1 | 1.831 | 14.907 | 15.665 | 0.084 |
| 0.5 | 0.949 | 1 | 0.961 | 0.305 | 1.300 | 3.593 |
| 0.5 | 1.096 | 1 | 1.718 | 1.598 | 2.623 | 1.621 |

**Table 2.** *Cont.*

| $T_s/L$ | $\tan(\varphi)\gamma'_s D_p/s_{um}$ | A | B | C | D | E |
|---|---|---|---|---|---|---|
| 0.7 | 0.754 | 0.302 | 0.015 | 0.203 | 3.422 | 4.655 |
| 0.7 | 0.848 | 0.602 | 0.321 | 0.161 | 1.794 | 4.572 |
| 0.7 | 0.949 | 0.874 | 0.869 | 0.314 | 1.448 | 3.357 |
| 0.7 | 1.096 | 1.063 | 1.709 | 0.889 | 1.862 | 3.471 |

### 4.3. Empirical Design Procedure and Validation

Equations (2)–(4) can be applied based on geometry parameters as $D_w D_p/L^2 = 0.05\sim0.1$ and soil properties with $s_{um} = 20\sim40$ kPa, $\varphi = 30°\sim40°$, which is in line with engineering practice [47–49]. The equations are calculated with $T_s/L = 0.1$, 0.3, 0.5 and 0.7, which can also be used for other values of $T_s/L$ through interpolating. The failure envelopes calculated by FEM and Equations (2)–(4) are compared in Figure 10b for an example, and good agreement is obtained. The maximum relative error of about 7% occurs when $T_s/L = 0.7$ and $V/(2\pi R^3 \gamma'_s)$ is relatively small, which means that the prediction of the equations can be considered conservative. Based on the above geometric dimensions and soil properties, a suggested design procedure is proposed: (i) determining the shape of the failure envelopes in the *V-H-M* domain with the assistance of Equation (2) and Table 2; (ii) calculating the absolute values of uniaxial lateral bearing capacity and vertical capacity using Equations (3) and (4) along with Tables 3 and 4; (iii) obtaining the lateral and moment bearing capacities under a certain vertical load of the composite foundation.

**Table 3.** Values of parameters related to $T_s/L$ in Equation (3) ($h_0$).

| $T_s/L$ | $a_1$ | $b_1$ | $c_1$ | $d_1$ | $e_1$ | $f_1$ | $g_1$ | $h_1$ |
|---|---|---|---|---|---|---|---|---|
| 0.1 | 1.61 | 230 | −0.21 | −2.78 | −1.22 | 399.44 | 0.19 | −12.22 |
| 0.3 | −15.99 | 457.22 | 0.53 | −11.67 | −16.99 | 957.22 | 1.33 | −37.78 |
| 0.5 | −2.25 | 164.44 | 0.07 | −2.22 | 0.09 | 948.33 | −0.035 | −25.56 |
| 0.7 | 7.52 | −260.56 | −0.6 | 15 | −9.02 | 1506.67 | 0.54 | −45 |

**Table 4.** Values of parameters related to $T_s/L$ in Equation (4) ($v_0$).

| $T_s/L$ | $a_2$ | $b_2$ | $c_2$ | $d_2$ |
|---|---|---|---|---|
| 0.1 | −72.57 | 1876.67 | −11.97 | 997.22 |
| 0.3 | −101.01 | 2206.11 | −21.93 | 2400 |
| 0.5 | −74.47 | 1263.89 | −3.82 | 3653.89 |
| 0.7 | −136.69 | 1598.33 | 42.76 | 4313.89 |

To validate the suitability and accuracy of the proposed empirical design procedure, it is used to estimate the bearing capacity of composite foundations reported by other researchers. Firstly, the centrifuge test data reported by Lehane et al. (2014) [27] on a composite foundation in pure sand are compared with the predicted results using the proposed method in this study ($D_w D_p/L^2 = 0.048$, $e/D_p = 8$, $V/V_{ult} = 0$). The lateral bearing capacities from the empirical formula and test results are 59.23 and 63.14 in normalized form, respectively, which indicates an error of only 6.61%. Then, to further validate the accuracy, the laboratory testing results from the centrifuge tests carried out by Stone et al. (2018) [36] are also compared with the prediction results using the proposed empirical formula in this work ($D_w D_p/L^2 = 0.094$, $e/D_p = 8$, $V/V_{ult} = 0$). The normalized lateral bearing capacities are 143.64 and 181.89 for the empirical design procedure and the centrifuge test, respectively. It reflects an error of 21.03%. The difference is probably because in pure sand, the shape of the failure envelope is slightly different to that of $T_s/L = 0.7$ in certain situations, and the empirical design procedure for the bearing capacities of the composite foundation in sand-over-clay deposit may not be perfectly suitable for that in single soil layer conditions. Since very limited results are available for composite

foundations in layered soil conditions, and the proposed method agrees well with the laboratory testing data in this study, further study with more testing should be carried out to validate the proposed method. Nevertheless, based on the above comparison with FEM results and the centrifuge testing data, it is believed that the proposed empirical formula could be employed with good accuracy.

## 5. Conclusions

In this study, numerical simulations are carried out on the composite foundation in sand-overlying-clay deposits under combined *V-H-M* loadings. Based on the parametric analyses, the bearing capacities and failure mechanisms are found to be closely related to the foundation geometry ($D_w/L$ and $L/D_p$), the soil properties ($\tan(\varphi)\gamma'_s D_p/s_{um}$), the upper sand layer thickness ($T_s/L$), the lateral loading height ($e/D_p = 8$), and the pre-vertical load ($V/V_{ult}$). The following conclusions can be drawn:

For the bearing capacities in the *V-H-M* domain, the foundation geometry has a more pronounced influence when the wheel diameter ($D_w/L$) exceeds 0.5. However, it makes little difference to the shape of the failure envelopes under combined *V-H-M* loadings. The sand layer thickness ($T_s/L$) influences the bearing capacity with a nonlinear manner and the trend is determined by the soil properties ($\tan(\varphi)\gamma'_s D_p/s_{um}$), which have non-negligible effects on the failure mechanism and the shape of failure envelopes in the *V-H-M* domain. Due to the P-Δ effect and the "punch-through" failure pattern, the vertical load ($V/V_{ult}$) influences the lateral bearing capacity with an optimal value when the maximum capacity occurs. Through imposing an additional bending moment onto the composite foundation, the lateral loading height ($e/D_p$) shows a negative effect on the bearing capacity and the stability of the foundation. In the end, an empirical design approach is proposed with Equations (2)–(4) and Tables 2–4 for the prediction of the composite foundation in sand-overlying clay deposit under the *V-H-M* combined loadings.

**Author Contributions:** Y.W. (conceptualization, formal analysis, methodology, validation, writing-original draft preparation); X.Z. (formal analysis, Funding acquisition, supervision, writing-review & editing); J.H. (writing-review & editing; validation). All authors have read and agreed to the published version of the manuscript.

**Funding:** This research was funded by the National Natural Science Foundation of China (Grant No. 51578231 & 52178329).

**Institutional Review Board Statement:** Not applicable.

**Informed Consent Statement:** Not applicable.

**Data Availability Statement:** Not applicable.

**Conflicts of Interest:** The authors declare no conflict of interest.

## Notations

| | |
|---|---|
| $\alpha$ | soil/pile friction coefficient |
| $D_w$ | diameter of friction wheel |
| $D_{w0}$ | dimensionless foundation geometry |
| $D_p$ | diameter of pile |
| $d_w$ | distance to the foundation center |
| $e$ | lateral loading height |
| $e_0$ | normalized lateral loading height |
| $E_c$ | Young's modulus of clay |
| $E_s$ | Young's modulus of sand |
| $H$ | lateral load |
| $H_{ult}$ | lateral capacity |
| $H_0$ | uniaxial horizontal capacity |
| $h_0$ | dimensionless uniaxial horizontal capacity |
| $H_{BD}$ | vertical boundary width |

| | |
|---|---|
| $K_0$ | horizontal earth pressure coefficient |
| $k$ | gradient of strength with depth |
| $L$ | embedment length of pile |
| $M$ | moment |
| $P_w$ | soil passive resistance under the wheel |
| $P_v$ | vertical pressure under the wheel |
| $P_p$ | soil passive resistance along the pile |
| $R$ | radius of the pile |
| $s_u$ | undrained shear strength of clay |
| $s_{um}$ | undrained shear strength of clay at the mudline |
| $T_w$ | thickness of friction wheel |
| $T_s$ | thickness of sand layer |
| $T_{collar}$ | thickness of wheel collar |
| $v_0$ | dimensionless vertical load |
| $V_{BD}$ | vertical boundary distance |
| $V$ | vertical load |
| $V_{ult}$ | vertical capacity |
| $\gamma'_c$ | effective unit weight of clay |
| $\gamma'_s$ | effective unit weight of sand |
| $z$ | depth below the mudline alongside the pile |
| $z_{max}$ | depth of maximum earth pressure |
| $\varphi$ | internal friction angle of sand fill |
| $\varphi_0$ | normalized internal friction angle of sand |
| $\psi$ | dilation angle |
| $\upsilon$ | Poisson's ratio |
| $A, B, C, D, E$ | coefficients of Equation (2) |
| $a_1, b_1, c_1, d_1, e_1, f_1, g_1, h_1$ | coefficients of Equation (3) |
| $a_2, b_2, c_2, d_2$ | coefficients of Equation (4) |

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
