# Peer review of "Bearing Capacity of Single Pile-Friction Wheel Composite Foundation on Sand-over-Clay Deposit under V-H-M Combined Loadings"

_applsci, doi:10.3390/app11209446_

Round 1

Reviewer 1 Report

Paper: Bearing Capacity of Single Pile-Friction Wheel Composite Foundation on Sand-over-Clay Deposit under V-H-M Combined Loadings

Authors: Yikang Wang, Xinjun Zou

In this study, an attempt has been made to investigate the effects of several factors on the bearing capacity and failure mechanism of the wind turbine foundation, i.e., single pile-friction wheel composite foundation. Wind turbines are important offshore structures related to green energy. Additionally, in order to extend the lifecycle of not only wind turbines but also other structures, the foundation plays a very important role. Therefore, this is an interesting topic, unfortunately, the manuscript does not reach to be published in the journal with its present form. In my opinion, a major revision needs to be performed.

I suggest revising the manuscript and answering the comments/questions as listed below. I also would be happy to read the article after revision to make sure that the authors had addressed all reviewer’s comments and suggestions. 

  1. I strongly concern about this comment: Relating directly to this topic, the author published at least 3 papers (but they were not cited in this manuscript) as follows:
  • Failure mechanism and lateral bearing capacity of monopile-friction wheel hybrid foundations in soft-over-stiff soil deposit, Marine Georesources & Geotechnology (2021)
  • Experimental study on the bearing capacity of large-diameter monopile in sand under water flow condition, Ocean Engineering (2021)
  • Experimental studies on the behaviour of single pile under combined vertical-torsional loads in layered soil, Applied Ocean Research (2021)
    • What is the main difference between this manuscript in comparison with three papers above?
    • Section 2.3: You conducted experiments for this structure. Why did not you verify the model using your experiments? If it is possible you also need to verify with your experiment.
    • Section 2.3 and 4.3: You used results from other previous studies (not your experiments) to verify and validate. I want to know that, almost all parameters such as loads, soil layer thickness, soil parameters, dimensions of pile and wheel, etc. in your model when compared with those works are similar or not? If they are much different, you cannot employ them to verify.

Additionally, please consider the comments below:

  1. The authors need to provide more detail about the load applied. For example, how did you select the value? Which standard did you follow to determine wind load, wave force, etc.?
  2. Lines 34-37: Please provide the references for this statement or give a short reason/explanation.
  3. Lines 43-53 and 57-59: The authors stated that “Most of these studies are only limited to the behaviors of these foundations in pure sand or clay” and “Sand-overlying-clay soil deposit is a complex but one of the most commonly encountered soil profiles in petroleum regions and offshore wind farms”. Is there any recommendation for the application of the Single Pile-Friction Wheel Composite Foundation related to the geological conditions?
  4. Section 1.2: The authors reviewed several previous works relating to your study, however, the difference between your and previous researches is not shown. Please indicate clearly the novelty compared to previous studies in this section (after line 93).
  5. In Line 98, Dw is the wheel thickness, however, Dw is the wheel diameter in line 134. Is this a mistake? Additionally, the authors employed many notations in the manuscript. I think it is better if a section named “notation” is added at the beginning or end of the manuscript.
  6. Lines 137-138: The authors provided that “The steel pile and wheel are modelled as an elastic-perfectly material”. What are the limitations related to results, recommendations from the model and application to real structures?
  7. The unit of many variables shown in the text part and equations of the manuscript are not indicated. Please check the entire manuscript and add them to suitable positions.

Reviewer 2 Report

Here are my remarks and comments on the work:

1.    The assessed work is interesting and carried out on the basis of a well-thought-out methodology. The authors applied the results of their research to the rich letter resources which, due to the composite nature of the solution, could not be complete, but constituted a solid basis for further analyses. Due to the fact that the bearing capacities and failure mechanisms of the considered composite foundation, which at the moment are not yet sufficiently recognized, have become the subject of analyses in this work, the presented results make it particularly valuable. The practical aspect of the work is to propose a design procedure that takes into account a number of leading parameters that have a direct impact on the final foundation solutions.

2.    In my opinion, the quality of the drawings will leave some doubts, eg Figure 1 or 2 (maybe enlargement would solve the problem?) and the descriptions (the legend on the right) of Figures 4,5,6 are completely illegible.

3.    The Tables are described inconsistently, e.g. the description of Tables 3 and 4 in line (342) is different (without brackets) than in line (382) Table (2) to (4) (in brackets). I recommend uniformity throughout the text.

4. Perhaps an even better solution, worth considering in the future, would be a pile with an extended base?

Round 2

Reviewer 1 Report

Thank you for your response

After reading the revised manuscript and the responses of the authors, I appreciate improvements in the revised manuscript. I found that almost all comments and questions were adequately responded, and corrections were made in the manuscript.

I recommend the manuscript for publication with its form now.